# Virtual mouse brain histology from multi-contrast MRI via deep learning

**Zifei Liang[1], Choong H Lee[1], Tanzil M Arefin[1], Zijun Dong[1], Piotr Walczak[2], Song-Hai Shi[3], Florian Knoll[1], Yulin Ge[1], Leslie Ying[4], Jiangyang Zhang[1]\***

[1]Bernard and Irene Schwartz Center for Biomedical Imaging, Department of Radiology, New York University School of Medicine, New York, United States; [2]Department of Diagnostic Radiology and Nuclear Medicine, University of Maryland, Baltimore, United States; [3]Developmental Biology Program, Sloan Kettering Institute, Memorial Sloan Kettering Cancer Center, New York, United States; [4]Departments of Biomedical Engineering, Electrical Engineering, University at Buffalo, the State University of New York, Buffalo, United States

**Abstract** [1]H MRI maps brain structure and function non-invasively through versatile contrasts that exploit inhomogeneity in tissue micro-environments. Inferring histopathological information from magnetic resonance imaging (MRI) findings, however, remains challenging due to absence of direct links between MRI signals and cellular structures. Here, we show that deep convolutional neural networks, developed using co-registered multi-contrast MRI and histological data of the mouse brain, can estimate histological staining intensity directly from MRI signals at each voxel. The results provide three-dimensional maps of axons and myelin with tissue contrasts that closely mimic target histology and enhanced sensitivity and specificity compared to conventional MRI markers. Furthermore, the relative contribution of each MRI contrast within the networks can be used to optimize multi-contrast MRI acquisition. We anticipate our method to be a starting point for translation of MRI results into easy-to-understand virtual histology for neurobiologists and provide resources for validating novel MRI techniques.

**\*For correspondence:**
jiangyang.zhang@nyulangone.org

**Competing interest:** The authors declare that no competing interests exist.

## Editor's evaluation

This paper demonstrates how MRI can be used to mimic histological measures. This is something that the field of MRI has dubbed virtual histology (or MR-histology) for a while, but this paper is the first convincing demonstration that it can be achieved.

## Introduction

Magnetic resonance imaging (MRI) is one of a few techniques that can image the brain non-invasively and without ionizing radiation, and this advantage is further augmented by a large collection of versatile tissue contrasts. While MRI provides unparalleled insight into brain structures and functions at the macroscopic level (*Lerch et al., 2017*), inferring the spatial organization of microscopic structures (e.g., axons and myelin) and their integrity from MR signals remains a challenging inverse problem. Without a thorough understanding of the link between MR signals and specific cellular structures, uncertainty often arises when determining the exact pathological events and their severities. The lack of specificity hinders direct translation of MRI findings into histopathology and limits its diagnostic value.

Tremendous efforts have been devoted to developing new mechanisms to amplify the affinity of MRI signals to target cellular structures in order to improve sensitivity and specificity. Recent progress

in multi-modal MRI promises enhanced specificity by integrating multiple MR contrasts that target distinct aspects of a cellular structure (*Mangeat et al., 2015*). For example, magnetization transfer (MT), $T_2$, and diffusion MRI are sensitive to the physical and chemical compositions of myelin, and combining them can lead to more specific myelin measurements than individual contrast (*Cercignani and Bouyagoub, 2018*). Progress in this front, however, has been hindered by the lack of realistic tissue models for inference and ground truth histological data for validation.

The objective of this study is to test whether deep convolutional neural networks (CNNs), developed using co-registered histology and MRI data, can bypass the above-mentioned obstacles and enhance our ability to map key cellular structures from MR signals. With its capability to bridge data acquired with different modalities (*Christiansen et al., 2018*; *Leynes et al., 2018*; *Ounkomol et al., 2018*), the deep learning framework (*LeCun et al., 2015*) has certain advantages over existing modeling approaches, as it is data-driven and not limited by particular models and associated assumptions. As MR signals are the ensemble average of all spins within each voxel, a typical set of three-dimensional (3D) MRI data, with millions of voxels, thus provides ample instances to train deep CNNs. Through training, the networks can potentially reconstruct the link between MR signals and cellular structures in co-registered histology and translate multi-contrast MRI data into maps that mimic histology. Our results demonstrate that this approach offers enhanced specificity for detecting axons and myelin compared to existing MRI-based markers. Furthermore, adding perturbations to the networks allows us to probe the relative contribution of individual MR contrast, which can be used to optimize multi-contrast MRI strategy and evaluate novel imaging contrasts.

## Results

### Prediction of auto-fluorescence images of the mouse brain from MR images using deep learning

We first demonstrated our method using co-registered 3D MRI and auto-fluorescence (AF) data. MRI dataset from ex vivo C57BL/6 mouse brain (P60, n = 6), each contained 67 3D MR ($T_2$, MT, and diffusion) images, were spatially normalized to the Allen Reference Atlas (ARA) (*Ng et al., 2009*; *Figure 1A*). We then selected 100 AF datasets from the Allen Mouse Brain Connectivity Atlas (AMBCA) (*Oh et al., 2014*) with minimal amounts of tracer signals in the forebrain. The contrast in the AF data is not specific to a particular structure, but a majority of hypo-intense regions co-localized with myelinated white matter tracts (*Christensen et al., 2014*). These 3D AF data had already been normalized to the ARA and were down-sampled to the resolution of the MRI data (0.06 mm isotropic). Mismatches between the MRI and AF images were mostly within one to two voxels (*Figure 1—figure supplement 1A-B*).

A deep CNN, named MRH-AF, which contained 30 residual blocks, was trained using multiple 3 × 3 patches from the forebrain region of each MRI data (40,000 patches, N = 6) as inputs and their corresponding patches in the co-registered AF data as targets (*Figure 1B*) (details on the network and training can be found in the Materials and methods section). In order to determine the amount of training data sufficient to capture the relationship between these two modalities, we performed separate training sessions with target AF data ranging from randomly selected 60 subjects (i), 6 subjects (ii), single subject data (iii), down to 5000 and 1000 3 × 3 patches randomly selected within a single subject data (iv and v) (*Figure 1C*). The 3 × 3 patch size was shown to accommodate residual mismatches between MRI and AF data (*Figure 1—figure supplement 1F-H*), and we chose such a small patch size instead of the entire image for training because we aimed to define the local relationship between cellular structures within an MRI voxel and corresponding ensemble-averaged MRI signals.

The performance of MRH-AF was evaluated using the average 3D AF data in the ARA (CCF version 3, average of 1675 mouse brains)(*Oh et al., 2014*) as the reference and MRI data from a separate group of mice (P60, n = 4) as the inputs. The MRH-AF results trained with 60-subject AF data as training targets (i) showed good agreement with the reference (*Figure 1D*) and strong voxel-wise signal correlation ($R^2$ = 0.71, p < 0.001, *Figure 1—figure supplement 3C*). The agreement was maintained for (ii) and (iii) both visually and quantitatively, as measured by the root mean square errors (RMSEs) and structural similarity index (SSIM) (*Figure 1D–E*). The specificity to hypo-intense regions in the reference defined by optimal thresholding was evaluated using receiver operating characteristic (ROC) analysis. The MRH-AFs trained with 60- and 6-subject AF data (i and ii) showed high specificity

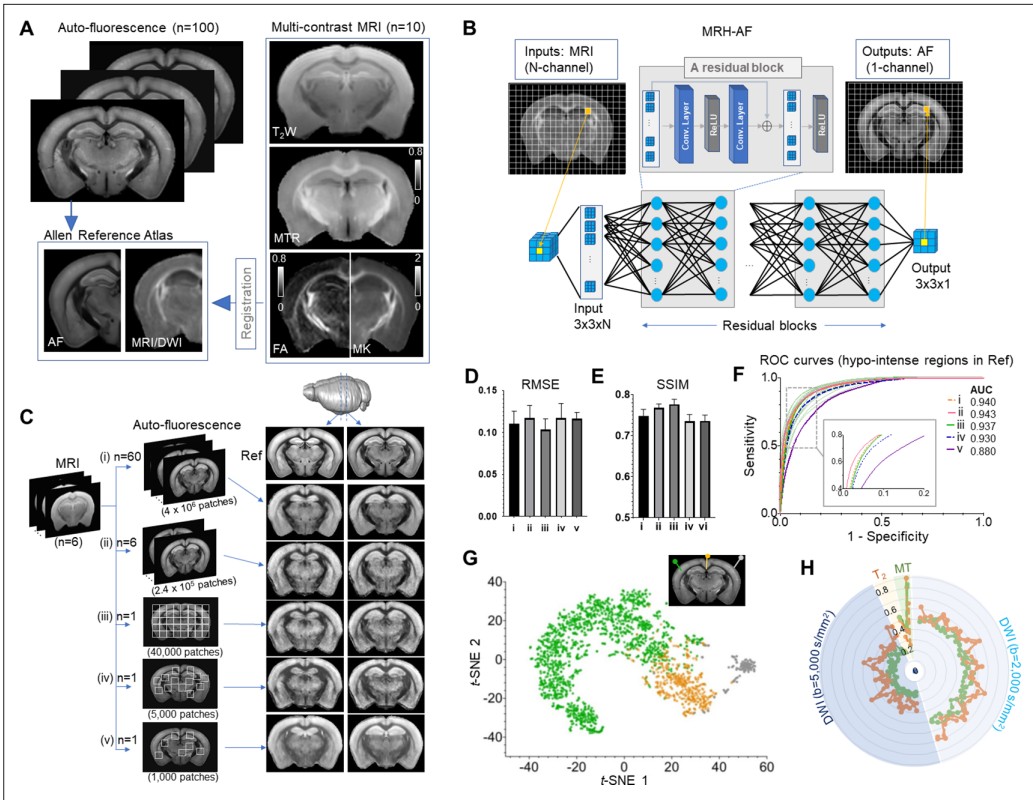

**Figure 1.** Connect multi-contrast magnetic resonance imaging (MRI) and auto-fluorescence (AF) data of the mouse brain using deep learning. (**A**) $T_2$-weighted ($T_2$W), magnetization transfer ratio (MTR), and diffusion-weighted images (DWIs) were registered to the Allen Reference Atlas (ARA) space, from which 100 already registered AF data were selected and down-sampled to the same resolution of the MRI data. Parameter maps derived from DWI, for example, fractional anisotropy (FA) and mean kurtosis (MK), were also included in this study. (**B**) The deep convolutional neural network (CNN) contained 64 layers (two layers for each residual block × 30 residual blocks plus four additional layers at the input and output ends) and was trained using multiple 3 × 3 MRI patches as inputs and corresponding 3 × 3 patches from histology as targets. (**C**) The CNN was trained using the MRI data (n = 6) and different amounts of randomly selected AF data (i–v). The results generated by applying the CNN to a separate set of MRI data (n = 4) were shown on the right for visual comparison with the reference (Ref: average AF data from 1675 subject). (**D–E**) Quantitative evaluation of the results in **C** with respect to the reference using root mean square error (RMSE) and structural similarity indices (SSIM). The error bars indicate the standard deviations due to random selections of AF data used to train the network. (**F**) The receiver operating characteristic (ROC) curves of the results in **C** in identifying hypo-intense structures in the reference and their areas under the curve (AUCs). The ROC curves from 25 separate experiments in (iii) (light green) show the variability with respect to the mean ROC curve (dark green) due to inter-subject variations in AF intensity. (**G**) The distribution of randomly selected 3 × 3 MRI patches in the network's two-dimensional (2D) feature space, defined using the t-SNE analysis based on case (iii) in **C**, shows three clusters of patches based on the intensity of their corresponding patches in the reference AF data (turquoise: hyper-intense, orange: hypo-intense; gray: brain surfaces). (**H**) MRI signals from two representative patches with hyper-intense AF signals (turquoise) and two patches with hypo-intense AF signals (orange). The orange profiles show higher DWI signals and larger oscillation among them than the turquoise profiles (both at b = 2000 and 5000 s/mm$^2$).

The online version of this article includes the following figure supplement(s) for figure 1:

**Figure supplement 1.** Evaluate the effects of mismatches between input magnetic resonance imaging (MRI) data and target auto-fluorescence (AF) data on deep learning outcomes.

**Figure supplement 2.** Training convergence curves of MRH auto-fluorescence (MRH-AF) network (**A–B**) and MRH myelin basic protein (MRH-MBP) during transfer learning.

**Figure supplement 3.** Evaluation of MRH auto-fluorescence (MRH-AF) results generated using modified 3 × 3 patches with nine voxels assigned the same values as the center voxel as inputs.

with areas under curve (AUCs) greater than 0.94, and the MRH-AF trained with 1-subject data (iii) had a slightly reduced average AUC of 0.937 (*Figure 1F*). The variation in the ROC curves in (iii), caused by the inter-subject variations in AF signals among subjects chosen for training, was relatively small. Further reducing the size of training data (iv and v) resulted in declined performances (*Figure 1D–F*), emphasizing the need for sufficient training data.

The way that MRH-AF in (iii) translated individual 3 × 3 MR patches into AF signals was visualized in a 2D feature space derived by t-distributed stochastic neighbor embedding (t-SNE) analysis (*van der Maaten and Hinton, 2008*; *Figure 1G*). Patches in the MRI data that were assigned with hypo-intense AF signals (orange) mostly clustered at the lower right corner, well separated from patches that were assigned with hyper-intense AF signals (turquoise) or near the brain surface (gray). Representative patches from the first two categories showed distinctive signal profiles (*Figure 1H*). Switching the network inputs to modified 3 × 3 patches, in which all voxels were assigned the same MR signals as the center voxel, resulted in no apparent loss in sensitivity and specificity and still produced strong voxel-wise signal correlation with the reference AF maps ($R^2$ = 0.68, p < 0.001, *Figure 1—figure supplement 3*), suggesting that MRH-AF primarily relied on multi-contrast MR signals, not patterns

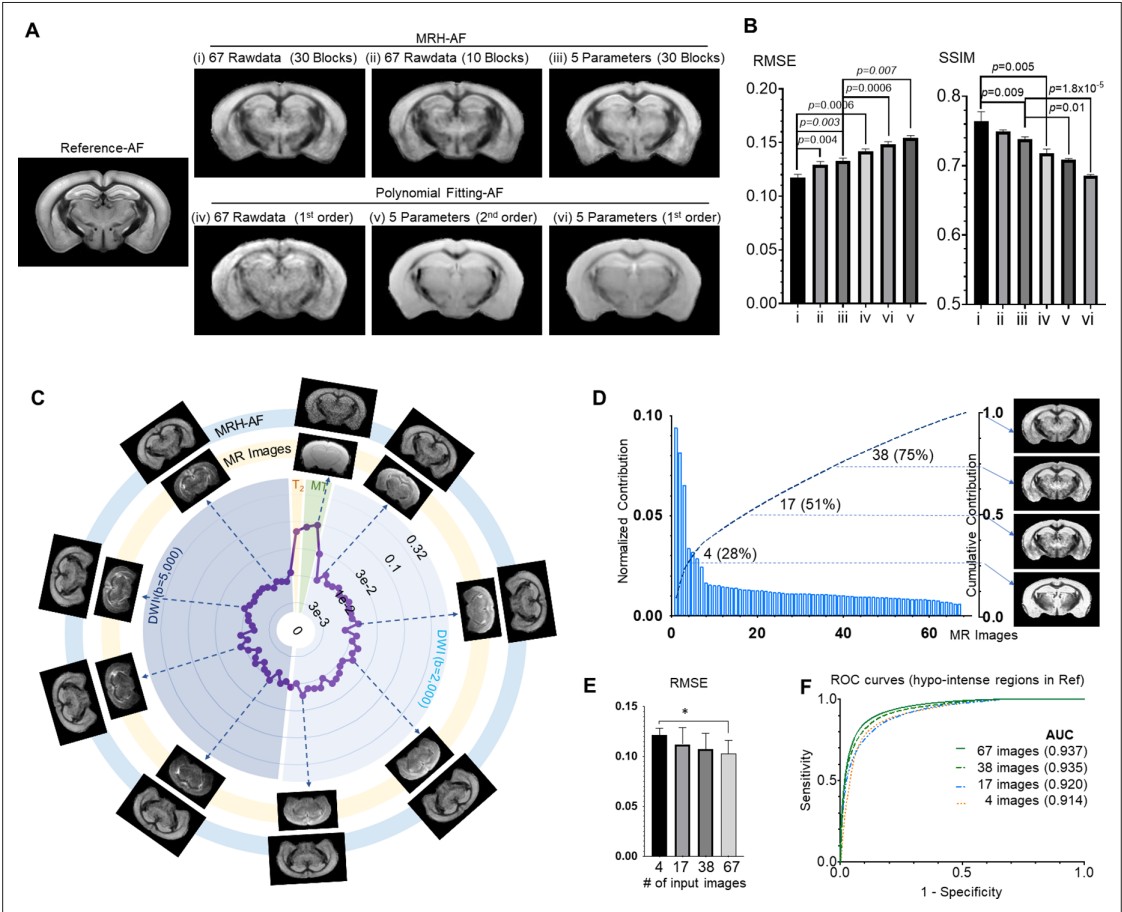

**Figure 2.** Understanding how multi-contrast magnetic resonance imaging (MRI) input influences the performance of MRH auto-fluorescence (MRH-AF). (**A**) MRH-AF results generated under different conditions (top panel) compared to polynomial fitting results (lower panel). (**B**) Root mean square error (RMSEs) and structural similarity indices (SSIMs) of the predicted AF maps shown in A with respect to the reference AF map. (**C**) Plots of the relative contribution of individual MRI images, normalized by the total contribution of all MR images, measured by RMSE. Images displayed on the outer ring (light blue, MRH-AF) show the network outcomes after adding 10% random noises to a specific MR image on the inner ring (light yellow). (**D**) The relative contributions of all 67 MR images arranged in descending order and their cumulative contribution. The images on the right show the MRH-AF results with the network trained using only the top 4, 17, 38, and all images as inputs. (**E**) RMSE measurements of images in D(n = 4) with respect to the reference AF data. Lower RMSE values indicate better image quality. * indicates statistically significant difference (p = 0.028, t-test). (**F**) Receiver operating characteristic (ROC) curves of MRH-AF results in D and the area under the curve (AUC) values.

The online version of this article includes the following figure supplement(s) for figure 2:

**Figure supplement 1.** Changes in network output after adding random noises to the original images.

within patches, to generate its predictions. Overall, the result demonstrates that the ability of MRH-AF to translate multi-contrast MRI data into maps that mimic the tissue AF contrast in the AMBCA.

Reducing the number of residual blocks from 30 to 10 resulted in a slight reduction in quality of the predicted AF map (*Figure 2A–B*) with significantly increased RMSEs but no significant change in SSIM (*Figure 2B*). MRH-AF outperformed first- and second-order polynomial fittings (*Figure 2A–B*, case i vs. case iv, case iii vs. cases v and vi), potentially due to the network's ability to accommodate remaining mismatch between AF and MRI data. Replacing the 67 rawdata with five commonly used MR parameter maps ($T_2$, MTR, FA, MD, and MK) as inputs to train the network produced less accurate AF predictions, for example, the loss of contrasts in the thalamus (*Figure 2A*) as well as significantly increased RMSEs and decreased SSIMs (*Figure 2B*). This suggests that the five MR parameters, although reflecting key tissue properties, do not contain all the information available from the 67 rawdata. However, using empirical or model-based MR parameters as inputs to the network has the advantage of broader applicability without requiring particular acquisition protocols and instruments, and a carefully selected and comprehensive set of such parameters will likely improve the predictions.

Based on the local ensemble average property of MR signals, we added random noises to each of the 67 MR images, one at a time, as perturbations to the network (*Olden et al., 2004*) and measured the effect on network outcomes with respect to noise-free results (*Figure 2C*), which reflected how each MR image influenced the outcome of MRH-AF or its relative contribution in the network. Similar information can also be obtained by training the networks with different subsets of the MRI contrasts and comparing the network predictions, but the perturbation method allows us to probe the existing network without retraining. We found that adding noises to a few images (e.g., $T_2$ and MT images) produced noticeably larger effects, in terms of output image quality and the ability of the network to separate different tissue types, than adding a comparable level of noises to other images (*Figure 2— figure supplement 1*), potentially due to redundant information shared among the 60 rawdata in the diffusion MRI dataset. This noise perturbation result can be used to accelerate MRI acquisition by prioritizing the acquisition of images or contrasts with high relative contributions. The top 4, 17, and 38 images ranked based on their contributions accounted for 28%, 50%, and 75% of the total contribution to the final result, respectively (*Figure 2D*). Results from training the network with the top 38 MR images as inputs showed comparable visual quality (*Figure 2D–E*) and diagnostic power (*Figure 2F*) as the results based on the full dataset, but only required 57% the imaging time.

## Use deep learning to generate virtual maps of axon and myelin and enhance specificity

Next, we trained our network using serial histological sections immuno-stained for neurofilament (NF) and myelin basic protein (MBP), two commonly used markers for axons and myelin, from the Allen mouse brain atlas. These images were down-sampled, normalized to the ARA (*Figure 3—figure supplement 1*). Part of the images were used for training and the rest were used as references. Due to limited histological images stained for myelin and axons, we adopted the transfer learning strategy (*Weiss et al., 2016*). Using the MRH-AF network as a starting point, we fixed most of its network layers while leaving the last three convolutional layers as trainable with MBP and NF-stained histological images.

The MRH results (*Figure 3A*) showed closer visual congruence with the histological references than commonly used MRI-based markers for axons (fractional anisotropy [FA]) and myelin (magnetization transfer ratio [MTR] and radial diffusivity [*Song et al., 2002*], $D_R$) as well as linear fitting results using the five parameters as in *Figure 2*. Even though MRH was trained using coronal sections, it can generate maps along other axes when applied to 3D MRI data (*Figure 3B*). The MRH-NF/MBP results also showed strong signal correlations with the reference data ($R^2$ = 0.61/0.73, respectively, *Figure 3—figure supplement 2*). ROC analyses (*Figure 3C*) on detecting axon and myelin-rich structures demonstrate improved specificity compared to any single MRI-based markers or 5-parameters linear fitting, while t-SNE analyses visualize how the two networks separate the patches in MRI data that corresponded to NF- and MBP-rich structures from the rest (*Figure 3D*).

The MRH results, in combination with the structural labels in ARA, provided insights into how the networks balanced multiple MRI contrasts to map axons and myelin in brain regions with distinct microstructural compositions. In the cortex, MRH-NF and MTR showed similar contrasts and comparable specificities to axons (*Figure 3E*), while in the whole brain, MTR had a noticeable lower specificity

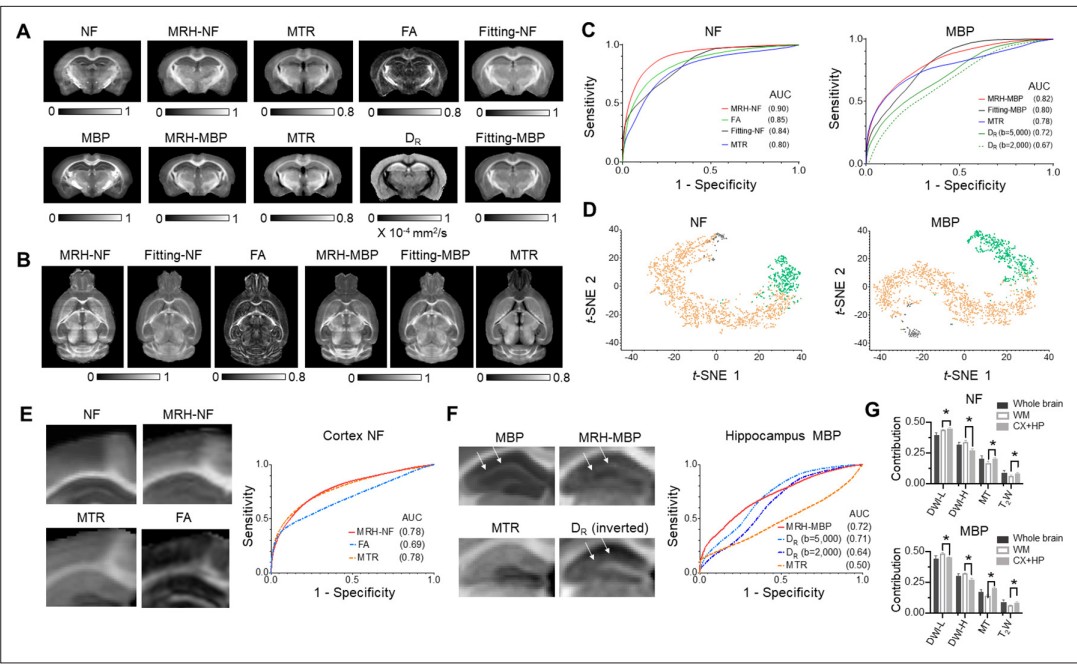

**Figure 3.** Inferring maps of neurofilament (NF) and myelin basic protein (MBP) from multi-contrast magnetic resonance imaging (MRI) data. (**A**) Comparisons of MRH-NF/MBP results with reference histology and MRI-based markers that are commonly used to characterize axon and myelin in the brain (MTR: magnetization transfer ratio; FA: fractional anisotropy; $D_R$: radial diffusivity) as well as linear prediction of NF and MBP (fitting-NF/MBP) based on five MRI parameter maps ($T_2$, MTR, FA, MD, and MK). (**B**) Even though MRH-NF/MBP were trained using coronal sections, they were able to generate maps for other orthogonal sections (e.g., horizontal sections shown here) from three-dimensional (3D) MRI, as expected from the local ensemble average property. The results show general agreements with structures in comparable horizontal MTR, FA, and five-parameter linear fitting maps. (**C**) Receiver operating characteristic (ROC) analyses of MRH-NF and MRH-MBP show enhanced specificity to their target structures defined in the reference data than MTR, FA, $D_R$, and five-parameter linear fittings. Here, $D_R$ values from diffusion-weighted images (DWIs) with b-values of 2000 and 5000 s/mm$^2$ are examined separately. (**D**) The distribution of randomly selected 3 × 3 MRI patches in the network's 2D feature spaces of MRH-NF and MRH-MBP defined using the t-distributed stochastic neighbor embedding (t-SNE) analyses. (**E–F**) Enlarged maps of the cortical (**E**) and hippocampal (**F**) regions of normal C57BL6 mouse brains comparing the tissue contrasts in MRH-NF/MBP with histology and MRI. In (**E**), white arrows point to a layer structure in the hippocampus. ROC analyses performed within the cortex and hippocampus show that MRH-NF/MBP have higher specificity than FA, MTR, and $D_R$, but with lower areas under the curve (AUCs) than in **C** due to distinct tissue properties. (**G**) Relative contributions of $T_2$-weighted ($T_2$W), MT, diffusion MRI (DWI-L: b = 2000 s/mm$^2$; DWI-H: b = 5000 s/mm$^2$) for the whole brain, white matter, and cortex/hippocampus. *: p < 0.005 (paired t-test, n = 4, from left to right, p = 0.0043/0.000021/0.00072/0.0014 for NF, p = 0.000058/0.000035/0.000002/0.00392 for MBP, respectively). Details on the contributions of each MRI contrast can be found in *Figure 3—figure supplement 3*.

The online version of this article includes the following figure supplement(s) for figure 3:

**Figure supplement 1.** Preparation and co-registration of serial two-dimensional (2D) histological sections to magnetic resonance imaging (MRI) data.

**Figure supplement 2.** Correlations between MRH generated results based on test magnetic resonance imaging (MRI) data and reference data for neurofilament (NF) and myelin basic protein (MBP).

**Figure supplement 3.** Plots of the contributions of 67 magnetic resonance (MR) images in MRH neurofilament (MRH-NF) (**A**) and MRH myelin basic protein (MRH-MBP) (**B**).

**Figure supplement 4.** Resolution of down-sampled auto-fluorescence (AF) reference data, MRH-AF data, and fractional anisotropy (FA) map of the input magnetic resonance imaging (MRI) data estimated using deconvolution analysis.

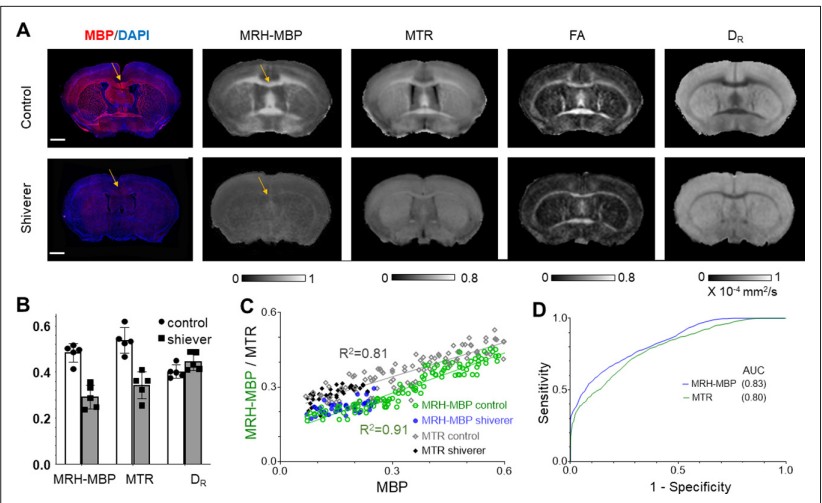

**Figure 4.** Comparisons of MRH myelin basic protein (MRH-MBP) with common magnetic resonance imaging (MRI)-based myelin markers in the shiverer mice. (**A**) Representative MRH-MBP results from dysmyelinated shiverer and control mouse brains show better agreement with histology than maps of magnetization transfer ratio (MTR), fractional anisotropy (FA), and $D_R$. (**B**) Differences in MRH-MBP, MTR, and $D_R$ values of the corpus callosum (t-test, n = 5 in each group, p = 0.00018/0.0061/0.475, respectively). (**C**) Voxel-wise analysis showed a slightly stronger correlation between MRH-MBP and actual MBP signals than MTR ($R^2$ = 0.91 vs. 0.81). (**D**) MRH-MBP showed slightly improved sensitivity and specificity for MBP-positive regions than MTR in the shiverer and control mouse brains. Scale bar = 1 mm.

than MRH-NF and FA (**Figure 3C**). This suggests that MRH-NF assigned additional weightings on MTR when processing cortical patches. Similarly, in ROC analysis for voxels within the hippocampus, the curve of MRH-MBP closely followed the curve of $D_R$ at b = 5000 s/mm², in a departure from the whole brain result (**Figure 3C**). Visual inspections of the MRH-MBP results revealed a layer structure in the hippocampus, which was not obvious in the MTR map but visible in the radial diffusivity ($D_R$) map at b = 5000 s/mm² (**Figure 3F**). Relative contributions of $T_2$-weighted ($T_2$W) and MT signals were significantly higher in the cortex and hippocampus than in white matter regions for both MRH-NF and MRH-MBP (**Figure 3G**).

Applying the MRH-MBP network to MRI data, collected from dysmyelinating *shiverer* and control mouse brains (n = 5/5) and not included in training MRH-MBP, generated maps that resembled the MBP-stained histology (**Figure 4A**). In the corpus callosum, the MRH-MBP results showed similar contrasts between *shiverer* and control moue brains as MTR (**Figure 4B**). Voxel-wise correlation between MRH-MBP predictions and co-registered MTR and MBP signals from the *shiverer* and control mice showed a slightly stronger correlation (**Figure 4C**) and small improvement in myelin specificity than MTR (**Figure 4D**).

## Use deep learning to generate maps that mimic Nissl staining

MRH networks can also be extended to other types of MR contrasts and histology. To demonstrate this, we used MRH to test whether cellularity in the mouse brain can be inferred from diffusion MRI signals, as our previous studies suggest that oscillating gradient spin echo (OGSE) (**Does et al., 2003**) diffusion MRI can generate a contrast similar to Nissl staining in both normal and injured mouse brains (**Aggarwal et al., 2014**; **Aggarwal et al., 2012**). We separated the down-sampled single subject 3D Nissl data from ARA into two parts. One was used as the training target, and the rest was used as the reference for testing (**Figure 5A**). The inputs to the so-called MRH-Nissl network included conventional pulsed gradient spin echo (PGSE) and recently developed OGSE diffusion MRI data. In the testing regions, the network that utilized all OGSE and PGSE data as inputs generated maps with good agreement with the ground truth Nissl data (**Figure 5A**), showing higher sensitivity and specificity than PGSE (**Figure 5B**). In the 2D feature space from t-SNE analysis (**Figure 5C**), the patches that correspond to regions with low Nissl signals were separated from other patches that correspond to regions with strong Nissl signals. Representative signal profiles from the three categories (**Figure 5D**)

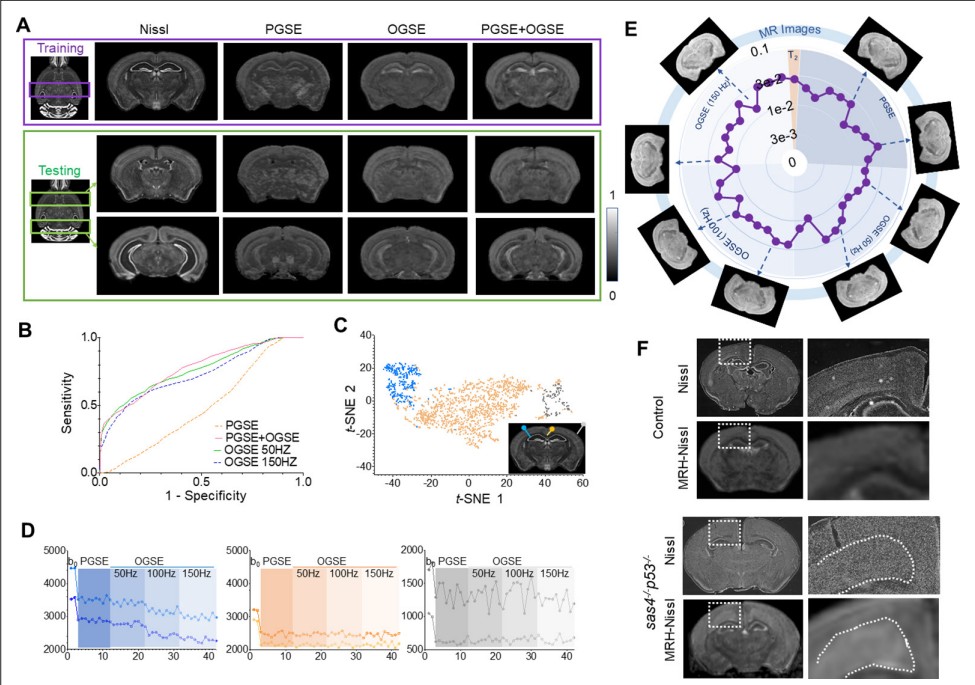

**Figure 5.** Generating maps that mimic Nissl stained histology from multi-contrast magnetic resonance imaging (MRI) data. (**A**) Comparisons of reference Nissl histology and MRH-Nissl results with pulsed gradient spin echo (PGSE), oscillating gradient spin echo (OGSE), combined PGSE and OGSE diffusion MRI data in both training and testing datasets. The entire datasets consist of PGSE and OGSE data acquired with oscillating frequencies of 50, 100, and 150 Hz, a total of 42 images. (**B**) Receiver operating characteristic (ROC) curves of MRH-Nissl show enhanced specificity for structures with high cellularity (strong Nissl staining) when both PGSE and OGSE data were included in the inputs than PGSE only. (**C**) The distribution of randomly selected 3 × 3 MRI patches in the network's 2D feature spaces of MRH-Nissl defined using t-distributed stochastic neighbor embedding (t-SNE) analyses. Green and orange dots correspond to regions with high and low cellularity, respectively, and gray dots represent patches on the brain surface. (**D**) Representative signal profiles from different groups in C. (**E**) Relative contributions of PGSE and three OGSE diffusion MRI datasets (**F**) Representative MRH-Nissl results from sas4[-/-] p53[-/-] and control mouse brains compared with Nissl-stained sections. The location of the cortical heterotopia, consists of undifferentiated neurons, is indicated by the dashed lines in the mutant mouse brain image.

revealed that signals in the high Nissl signal patches decreased as the oscillating frequency increased, whereas the other two types of patches showed no such pattern. Detailed analysis of contrast contribution showed that PGSE and OGSE data contribute equally (*Figure 5E*), indicating the importance of OGSE data in generating the target tissue contrast. The MRH-Nissl map of the *sas4[-/-]p53[-/-]* mouse brain, which contains a band of heterotopia consists of undifferentiated neurons (*Insolera et al., 2014*), produced image contrasts that matched Nissl-stained histology (*Figure 5F*).

## Discussion

The present study focused on inferring maps of key cellular structures in the mouse brain from multi-contrast MRI data. Previous works on this problem include: new MRI contrasts that capture specific aspects of cellular structures of interest (*Stikov et al., 2015*; *Veraart et al., 2020*); carefully constructed tissue models for MR signals (*Jelescu and Budde, 2017*); statistical methods to extract relevant information from multi-contrast MRI (*Mangeat et al., 2015*); and techniques to register histology and MRI data (*Tward et al., 2020*; *Xiong et al., 2018*) for validation (*Schilling et al., 2018*; *Stolp et al., 2018*). Here, we built on these efforts by demonstrating that deep learning networks trained by co-registered histological and MRI data can improve our ability to detect target cellular structures.

Previous studies on the relationship between histology and MRI signals focused on correlating histological and MRI markers as co-registered MRI and histological data as well as realistic tissue models are scarce (*Jelescu and Budde, 2017*; *Novikov et al., 2018*). Adopting similar approaches

described by recent reports (*Christiansen et al., 2018*; *Ounkomol et al., 2018*) on using deep learning to generate histological labels from unlabeled light microscopy images, we demonstrate a proof of concept of using deep learning to solve the inverse problem of inferring histological information from MRI signals. Even though the resolution of the virtual histology is inevitably limited by the resolution of the input MRI data (~100 μm/voxel) (*Figure 3—figure supplement 4*), the presented approach has many potential applications in biomedical research involving MRI. It can enhance our ability to accurately map selected cellular structures and their pathology in mouse models of diseases using non-invasive MRI, with contrasts familiar to neurobiologists. Although the networks cannot be applied to human MRI directly due to vast differences in tissue properties and scanning protocols, understanding how the networks improve specificity based on given MRI contrasts will guide the development of optimal imaging strategy in the clinics. In addition, the co-registered histology and MRI dataset provide a testbed for developing new MRI strategies. As it is relatively easy to normalize any new MRI data to our 3D multi-contrast MRI data and co-registered histology, the sensitivity and specificity of a new MRI contrast to target cellular structures can be evaluated. With quantitative information on the contributions of different MR contrasts, it is now straightforward to design accurate and efficient multi-contrast MRI strategy.

Perfect co-registration between MRI and histology is highly challenging, as conventional tissue preparations inevitably introduce tissue deformation and damages. In addition, differences in tissue contrasts between histology and MRI also limit the accuracy of registration. Serial two-photon tomography used by the Allen Institute and similar methods allow 3D uniform sampling of the entire brain, which facilitate registration using established registration pipelines (*Kuan et al., 2015*). We expect recent advances in tissue clearing techniques can assist in this aspect once issues such as tissue shrinkage and penetration of antibodies for more target cellular structures are resolved. Remaining mismatches can be accommodated by choosing the appropriate patch size in the network as shown in our results and earlier studies (*Rivenson et al., 2019*).

There are several directions to further improve our work. First, it is important to curate a training dataset that covers a broad spectrum of normal and pathological conditions and, ideally, with MRI and histology data acquired from the same animal. The data included in this study were adult mouse brain and most white matter structures are myelinated. As a result, the network to predict axons place a substantial weight on MRI contrasts that reflect myelin content (e.g., MT). With the inclusion of unmyelinated embryonic or neonatal mouse brains, we anticipate that the contribution of myelin will be reduced. Inclusion of pathological examples, such as *shiverer* mouse brain data, for training will likely improve our ability to characterize pathological conditions. The result from the *shiverer* data, while demonstrating the usage of the tool, does not tell us whether it will work for cases with more complex pathology, which will require further investigation. Second, the CNNs constructed in this study involved several common building blocks of deep learning, and new advances on network architecture design (e.g., *Goodfellow et al., 2014*; *Zhu et al., 2017*) could further enhance the performance. While CNNs have been commonly treated as black boxes, several recently reported approaches, such as deep Taylor decomposition (*Montavon et al., 2017*) and Grad-CAM (*Selvaraju et al., 2017*), can help explain the inner working. Third, developing similar networks for in vivo MRI data and potential clinical application will require additional effort. The MRI data used in this study were collected from post-mortem mouse brain specimens, which are different from in vivo mouse brains due to death and chemical fixation. Differences in tissue properties between human and mouse brains also require additional steps. MRI parameters, either empirical or model based, may help to bridge this gap as they are translatable and less dependent on acquisition settings. As demonstrated in *Figure 2*, networks trained using well-selected MRI parameters can predict tissue histology with reasonable accuracy. Finally, deep learning cannot replace the good understanding of the physics involved in MRI contrasts and the development of new MRI contrast that targets specific cellular structures.

## Materials and methods
### Animals and ex vivo MRI
All animal experiments have been approved by the Institute Animal Care and Use Committee at New York University, Memorial Sloan Kettering Cancer Center, and Johns Hopkins University. Adult C57BL/6 mice (P60, n = 10, Charles River, Wilmington, MA), *sas4*$^{-/-}$*p53*$^{-/-}$ (*Insolera et al., 2014*) and

littermate controls (n = 4/4, P28), *rag2*⁻/⁻ *shiverer* and littermate controls (n = 5/5, P50) were perfusion fixed with 4% paraformaldehyde (PFA) in PBS. The samples were preserved in 4% PFA for 24 hr before transferring to PBS. Ex vivo MRI of mouse brain specimens was performed on a horizontal 7 T MR scanner (Bruker Biospin, Billerica, MA) with a triple-axis gradient system. Images were acquired using a quadrature volume excitation coil (72 mm inner diameter) and a receive-only four-channel phased array cryogenic coil. The specimens were imaged with the skull intact and placed in a syringe filled with Fomblin (perfluorinated polyether, Solvay Specialty Polymers USA, LLC, Alpharetta, GA) to prevent tissue dehydration (*Arefin et al., 2021*). Three-dimensional diffusion MRI data were acquired using a modified 3D diffusion-weighted gradient- and spin-echo sequence (*Wu et al., 2013*) with the following parameters: echo time (TE)/repetition time (TR) = 30/400 ms; two signal averages; field of view (FOV) = 12.8 mm × 10 mm × 18 mm, resolution = 0.1 mm × 0.1 mm × 0.1 mm; two non-DWIs ($b_0$s); 30 diffusion encoding directions; and b = 2000 and 5000 s/mm$^2$, total 60 diffusion-weighted images (DWIs). Co-registered $T_2$W and MT MRI data were acquired using a rapid acquisition with relaxation enhancement sequence with the same FOV, resolution, and signal averages as the diffusion MRI acquisition and the following parameters: $T_2$: TE/TR = 50/3000 ms; MT: TE/TR = 8/800 ms, one baseline non-MT-weighted ($M_0$) image and one MT-weighted ($M_t$) images with offset frequency/power = –3 KHz/20 µT were acquired. The total imaging time was approximately 12 hr for each specimen. For the *sas4*⁻/⁻*p53*⁻/⁻ and littermate controls (n = 4/4, P28), PGSE and OGSE diffusion MRI data were acquired with the protocol described in *Aggarwal et al., 2012* and a spatial resolution of 0.1 mm × 0.1 mm × 0.1 mm. All 3D MRI data were interpolated to a numerical resolution of 0.06 mm × 0.06 mm × 0.06 mm to match the resolution of our MRI-based atlas (*Chuang et al., 2011*).

MTR images were generated as MTR=($M_0$–$M_t$)/$M_0$. From the diffusion MRI data, diffusion tensors were calculated using the log-linear fitting method implemented in MRtrix (http://www.mrtrix.org) at each voxel, and maps of mean and radial diffusivities and FA were generated, The mouse brain images were spatial normalized to an ex vivo MRI template (*Chuang et al., 2011*) using the large deformation diffeomorphic metric mapping (LDDMM) method (*Ceritoglu et al., 2009*) implemented in the DiffeoMap software (https://www.mristudio.org). The template images had been normalized to the ARA using landmark-based image mapping and LDDMM.

## Histological data

From the Allen moue brain atlas, single subject 3D Nissl data and 3D AF data (n = 100), which were already registered to the ARA space, were down-sampled to 0.06 mm isotropic resolution. The *sas4*⁻/⁻*p53*⁻/⁻, *rag2*⁻/⁻ *shiverer* and control mouse brains were cryopreserved and cut into 30 µm coronal sections and processed for Nissl and immunofluorescence. For immunofluorescence, sections were first washed with PBS, blocked with 5% bovine serum albumin, and incubated overnight at 4°C with primary antibodies: anti-MBP (AbD Serotec, MCA4095). Sections were rinsed with PBS and incubated with Alexa Fluor secondary antibodies (Invitrogen) cover-slipped with anti-fade mounting medium containing DAPI (Vectrolabs, H-1200). Images were obtained and tile-stitched using an inverted microscope (Zeiss, Axio Observer.Z1) equipped with a motorized table.

## Registration of MRI and histological data

Group average 3D MRI data in our previously published mouse brain atlas (*Chuang et al., 2011*) were first spatially normalized to the ARA space. Briefly, 14 major brain structures (e.g., cortex, hippocampus, striatum) in the atlas MRI data were manually segmented following the structural delineations in the ARA. Voxels that belong to these structures in the MRI and average 3D AF data in the ARA (down-sampled to 0.06 mm isotropic resolution) were assigned distinct intensity values, and a diffeomorphic mapping between the discretized atlas MRI and ARA AF data was computed using LDDMM. The mapping was then applied to the original atlas MRI data to generate an MRI template registered to the ARA space. Using dual-channel LDDMM (*Ceritoglu et al., 2009*) based on tissue contrasts in the average DWI and FA images and the MRI template, the 3D MRI data acquired in this study were accurately normalized to the ARA space.

NF- and MBP-stained images of the C57BL/6 mouse brain were downloaded from the ARA reference dataset. Images with major artifacts or tissue loss were excluded. Small tissue tearing and staining artifacts were removed using the inpainting feature implemented in the photoshop heading brush tool (https://www.adobe.com), and dark voxels in the ventricles were replaced by the average

intensity values of the cortex to match MRI data (*Figure 3—figure supplement 1*). The repaired images were down-sampled to an in-plane resolution of 0.06 mm/voxel. For each 2D histological image, the best-matching MRI section in the MRI template was identified, and a coarse-to-fine alignment from histology to MRI using affine transform and landmark-based image warping tool in ImageJ (https://imageJ.nete/BUnwarpJ). The aligned 2D sections were then assembled into a 3D volume and mapped to the MRI template using LDDMM (between NF/MBP and FA) to further improve the quality of registration.

## Evaluation of image resolution

The resolution of MRI, histological images, were evaluated using a parameter-free decorrelation analysis method (*Descloux et al., 2019*), without the initial edge apodization step.

## Design and training of the MRH networks

MRH networks are constructed using a CNN model with convolutions from the MRI to histology space. The networks were implemented using the deep learning toolbox in Matlab (https://www.mathworks.com) using the directed acyclic graph architecture. To accommodate residual mismatches between MRI and histological data, the network consistently applied convolutional layers until the end layer that computed the distance loss to the target histology. The number of layers and neurons in each layer was determined empirically to balance performance and the risk of overfitting. We chose 64 hidden layers (60 layers in 30 residual blocks plus four layers at the input and output ends), each with 64 neurons, which applied a filter size of 3 × 3 and included a rectified linear unit (ReLu). Most of the hidden layers utilized skip connections to jump over layers to avoid vanishing gradients (*He et al., 2016*). The network was initialized with orthogonal random weights and was trained using a stochastic gradient descent optimizer to minimize voxel-level absolute distance loss (*Figure 1—figure supplement 2A-B*). Several choices of mini batch sizes were tested and the size was set to 128 to attain a balance between training speed and performance. Stochastic gradient descent with a momentum beta of 0.9 was used for stochastic optimization, with an initial learning rate of 0.1 and a learning rate factor of 0.1. During training, the learning rate was reduced by a factor of 0.1 every 10 epochs. Maximum epoch number was set at 60, but early stopping was employed if the validation set loss did not decrease in five epochs. During hyper-parameter tuning, 1000 3 × 3 patches were randomly held out to as the validation dataset and isolated from the training dataset. The weights from the epoch with the lowest validation loss were selected for final testing.

When retraining the MRH-AF neural network using MBP/NF data, we refined the last three layers' parameters while leaving the parameters in other layers untouched in the Matlab deep learning toolbox using the directed acyclic graph architecture. The hyperparameters and training patches are the same as MRH-AF. Specifically, the network training initial learning rate was 0.0001, while learning rate factor was 0.1 to accomplish transfer learning. Using the stochastic gradient descent optimizer, our transfer learning converged as shown in *Figure 1—figure supplement 2C-D*.

## t-SNE analysis

The t-SNE cluster was performed using the Matlab t-SNE analysis function on the network prediction based on values in 2000 randomly selected 3 × 3 patches in the mouse brain MRI data.

## Contribution analysis

Following the perturbation method described by *Olden et al., 2004*, Rician noises were added to one input MR image to the pre-trained MRH networks, and RMSE between the noise contaminated outputs and the original output without noise was recorded. By repeating this procedure for all MR images, the sensitivities of MRH to each of the 67 input MR image or their contributions were obtained.

## Evaluate the effect of voxel mismatches

In the experiment that used the DWI and FA data of the mouse brains to train an MRH network, simulated voxel displacements were used to deform the FA data (target), which were perfectly co-registered to the DWI data (inputs), to test the effect of voxel mismatches between input and target data on network prediction. Gaussian random displacement fields were generated for pixels on a 1 mm by 1 mm grid in the coronal plane and propagated to other voxels by B-spline interpolations. The

displacement fields followed a Chi distribution with 2 degrees of freedom and were adjusted to match the level of voxel mismatches observed between MRI and histological data.

## Statistical analysis

Statistical significance was determined using unpaired Student's t-test with threshold set at 0.05. All statistical tests were performed with Prism (GraphPad). All values in bar graphs indicate mean + standard deviation.

## Acknowledgements

This work was supported by NIH grants R01NS102904 and R01HD074593.

## Additional information

### Funding

| Funder | Grant reference number | Author |
| --- | --- | --- |
| Eunice Kennedy Shriver National Institute of Child Health and Human Development | R01HD074593 | Jiangyang Zhang |
| National Institute of Neurological Disorders and Stroke | R01NS102904 | Jiangyang Zhang |

The funders had no role in study design, data collection and interpretation, or the decision to submit the work for publication.

### Author contributions

Zifei Liang, Conceptualization, Data curation, Formal analysis, Investigation, Methodology, Project administration, Resources, Software, Validation, Visualization, Writing – original draft, Writing - review and editing; Choong H Lee, Data curation, Methodology; Tanzil M Arefin, Data curation, Formal analysis; Zijun Dong, Formal analysis, Writing – original draft; Piotr Walczak, Data curation, Investigation, Methodology, Writing – original draft; Song-Hai Shi, Data curation, Writing – original draft; Florian Knoll, Formal analysis, Investigation, Methodology, Validation, Writing – original draft; Yulin Ge, Conceptualization, Funding acquisition, Writing – original draft; Leslie Ying, Conceptualization, Funding acquisition, Methodology, Writing – original draft; Jiangyang Zhang, Conceptualization, Data curation, Formal analysis, Funding acquisition, Investigation, Methodology, Project administration, Resources, Software, Supervision, Validation, Visualization, Writing – original draft, Writing - review and editing

### Author ORCIDs

Jiangyang Zhang (ID) http://orcid.org/0000-0003-0740-2662

### Ethics

This study was performed in strict accordance with the recommendations in the Guide for the Care and Use of Laboratory Animals of the National Institutes of Health. All of the animals were handled according to approved institutional animal care and use committee (IACUC) protocols (s16-00145-133) of the New York University.

### Decision letter and Author response

Decision letter https://doi.org/10.7554/eLife.72331.sa1
Author response https://doi.org/10.7554/eLife.72331.sa2

## Additional files

### Supplementary files
• Transparent reporting form

### Data availability

All data and source codes used in this study are available at https://www.github.com/liangzifei/MRH-net/ (copy archived at swh:1:rev:f116deb1fa6eedde6fc4aa4c5b6edf72a88d058d). The data can also be found at https://doi.org/10.5061/dryad.1vhhmgqv8.

The following dataset was generated:

| Author(s) | Year | Dataset title | Dataset URL | Database and Identifier |
|---|---|---|---|---|
| Liang Z, Zhang J | 2022 | Data fromMulti-contrast MRI and histology datasets used to train and validate MRH networks to generate virtual mouse brain histology | https://doi.org/10.5061/dryad.1vhhmgqv8 | Dryad Digital Repository, 10.5061/dryad.1vhhmgqv8 |

The following previously published dataset was used:

| Author(s) | Year | Dataset title | Dataset URL | Database and Identifier |
|---|---|---|---|---|
| Lein ES | 2006 | Reference data | http://connectivity.brain-map.org/static/referencedata | Allen Mouse Brain Atlas, referencedata |

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
