## [Editor Report]

This paper demonstrates how MRI can be used to mimic histological measures. This is something that the field of MRI has dubbed virtual histology (or MR-histology) for a while, but this paper is the first convincing demonstration that it can be achieved.

---

## [Decision Letter]

**Decision letter after peer review:**

Thank you for submitting your article "Virtual Mouse Brain Histology from Multi-contrast MRI via Deep Learning" for consideration by *eLife*. Your article has been reviewed by 2 peer reviewers, including Saad Jbabdi as Reviewing Editor and Reviewer #1, and the evaluation has been overseen by Floris de Lange as the Senior Editor.

Essential revisions:

1) As per reviewer #2, please include some quantitative evaluations of the model predictions in the cases where MRI and histology are available in the same animals (e.g. in the mutant mice).

2) Please make sure to address reviewer #1's points on the added value of a deep net , the concerns on the generalisability of the method, and the validity of the noise contamination experiments.

*Reviewer #2 (Recommendations for the authors):*

I would advocate two additional sets of experiments; in the simple form, align the existing histology maps (from, for example, the shiverer mice) to the MRIs from the same animals to provide direct quantifiable estimates of where the predicted histology slices deviate from the actual histology slices.

More broadly, this paper would greatly benefit from an enhanced training set consisting of MRI and histology from the same animals co-aligned. This would both provide better training and, importantly, better evaluation of the trained maps.

---

## [Author Response]

Essential revisions:1) As per reviewer #2, please include some quantitative evaluations of the model predictions in the cases where MRI and histology are available in the same animals (e.g. in the mutant mice).

We have evaluated the performance of MRH-MBP using MRI and myelin histology (MBP) data from shiverer and control mouse brains (Figure 4), which showed a stronger correlation between MRH-MBP predictions and actual MBP signals as well as higher myelin specificity than magnetization transfer ratio (MTR), a commonly used marker for myelin. We have also clarified that the shiverer and control data were not used for training the MRH-MBP network.

2) Please make sure to address reviewer #1's points on the added value of a deep net , the concerns on the generalisability of the method, and the validity of the noise contamination experiments.

We have revised our manuscript based on rev1’s comments. Specifically,

1) We have included results from a network trained using five summary measures as inputs (Figure 2A-B). The results suggest that using summary measures as inputs to the network can generate reasonable predictions of histology, which improve the prospect of generalizing the method to data generated from other MR systems.

2) We have compared the deep net with linear and quadratic fitting (Figure 2A-B), and the result demonstrates that deep learning network outperforms linear/quadratic fitting. One reason for this difference is that the deep network can accommodate residual mismatches between histology and MRI data (as demonstrated in Figure 1 —figure supplement 1). Such mismatches, often hard to completely remove, will affect polynomial fitting.

3) We also examined the need for deep net by varying the number of layers and compared the results. As shown in Figure 2A, lowering the number of residual blocks, each consists of two layers, from 30 to 10, still produce reasonable predictions for tissue auto-fluorescence (AF) signals. Further reducing the number of layers produces less satisfactory results. Furthermore, the number of layers needed also depends on the target histology. While a network with 10 residual blocks (~ 24 layers) can generate satisfactory predictions for tissue auto-fluorescence, 30 residual blocks (~64 layers) are needed to generate satisfactory predictions for Nissl staining as shown in Figure 5. We also clarified that convolution mostly work across channels in our network.

4) We have clarified the noise perturbation experiments. In the revised manuscript, we have removed the supplementary figure on t-SNE analysis of the noise perturbation, clarified that there are redundant information in the diffusion MRI dataset. We have focused on using the information to accelerate image acquisition as shown in Figure 2.

Reviewer #2 (Recommendations for the authors):I would advocate two additional sets of experiments; in the simple form, align the existing histology maps (from, for example, the shiverer mice) to the MRIs from the same animals to provide direct quantifiable estimates of where the predicted histology slices deviate from the actual histology slices.

We have performed the experiment comparing predicted myelin histology from shiverer and control mice, whose data were not used to train MRH-MBP, with MBP stained histology from the same animals. The result in Figure 4 shows that the predicted MBP signals correlated strongly with the actual MBP signals (R^2^=0.91) and have higher sensitivity and specificity than magnetization transfer ratio (MTR), a commonly used MRI marker for myelin. However, in the Discussion section, we emphasized that more work needs to be done to investigate whether the tool works for more complex myelin pathology.

More broadly, this paper would greatly benefit from an enhanced training set consisting of MRI and histology from the same animals co-aligned. This would both provide better training and, importantly, better evaluation of the trained maps.

We agree with the reviewer on this point. Currently, it is still a time consuming and challenging task to acquire high quality 3D histology from tissue specimens. Although tissue clearing promising techniques have made significant progress over the recent years, getting antibodies into deep structures in the adult mouse brain remains inconsistent. We have just started working with two groups to collect co-registered histology and MRI data, and it may take a year to fine-tune the protocols to obtain high quality 3D histology and co-registration process.